

# Machine Learning and Committee Models for Improving ECMWF Subseasonal to Seasonal (S2S) Precipitation Forecast

Mohamed Elneel Elshaikh Eltayeb Elbasheer[1,★], Gerald Augusto Corzo[1,★], Dimitri Solomatine[1] and Emmanouil Varouchakis[2]

[1]Hydroinformatics department, IHE Delft Institute for Water Education, Delft, Netherlands
[2]Technical University of Crete, Crete, Greece
★These authors contributed equally to this work.

*Correspondence to*: Mohamed Elneel Elshaikh Eltayeb Elbasheer (m.elbasheer@outlook.com)
         Gerald Augusto Corzo (g.corzo@un-ihe.org)
         Dimitri Solomatine (d.solomatine@un-ihe.org)
         Emmanouil Varouchakis (evarouchakis@gmail.com)

**Abstract.** The European Centre for Medium-Range Weather Forecasts (ECMWF) provides subseasonal to seasonal (S2S) precipitation forecasts; S2S forecasts extend from two weeks to two months ahead; however, the accuracy of S2S precipitation forecasting is still underdeveloped, and a lot of research and competitions have been proposed to study how machine learning (ML) can be used to improve forecast performance. This research explores the use of machine learning techniques to improve the ECMWF S2S precipitation forecast, here following the AI competition guidelines proposed by the S2S project and the World Meteorological Organisation (WMO). A baseline analysis of the ECMWF S2S precipitation hindcasts (2000–2019) targeting three categories (above normal, near normal and below normal) was performed using the ranked probability skill score (RPSS) and the receiver operating characteristic curve (ROC). A regional analysis of a time series was done to group similar (correlated) hydrometeorological time series variables. Three regions were finally selected based on their spatial and temporal correlations. The methodology first replicated the performance of the ECMWF forecast data available and used it as a reference for the experiments (baseline analysis). Two approaches were followed to build categorical classification correction models: (1) using ML and (2) using a committee model. The aim of both was to correct the categorical classifications (above normal, near normal and below normal) of the ECMWF S2S precipitation forecast. In the first approach, the ensemble mean was used as the input, and five ML techniques were trained and compared: k-nearest neighbours (k-NN), logistic regression (LR), artificial neural network multilayer perceptron (ANN-MLP), random forest (RF) and long–short-term memory (LSTM). Here, we have proposed a gridded spatial and temporal correlation analysis (autocorrelation, cross-correlation and semivariogram) for the input variable selection, allowing us to explore neighbours' time series and their lags as inputs. These results provided the final data sets that were used for the training and validation of the machine learning models. The total precipitation (tp), two-metre temperature (t2m) and time series with a resolution of 1.5 by 1.5 degrees were the main variables used, and these two variables were provided as the global ECMWF S2S real-time forecasts, ECMWF S2S reforecasts/hindcasts and observation data from the National Oceanic and Atmospheric



Administration (Climate Prediction Centre, CPC). The forecasting skills of the ML models were compared against a reference model (ECMWF S2S precipitation hindcasts and climatology) using RPSS, and the results from the first approach

showed that LR and MLP were the best ML models in terms of RPSS values. In addition, a positive RPSS value with respect to climatology was obtained using MLP. It is important to highlight that LSTM models performed quite similarly to MLP yet had slightly lower scores overall. In the second approach, the committee model (CM) was used, in which, instead of using one ECMWF hindcast (ensemble mean), the problem is divided into many ANN-MLP models (train each ensemble member independently) that are later combined in a smart ensemble model (trained with LR). The cross-validation and testing of the

CMs showed positive RPSS values regarding climatology, which can be interpreted as improved ECMWF on the three climatological regions. In conclusion, ML models have very low—if any—improvement, but by using a CM, the RPSS values are all better than the reference forecast. This study was done only on random samples over three global regions; a more comprehensive study should be performed to explore the whole range of possibilities.

## 1    Introduction

The S2S forecast is defined within the range of two weeks to two months. This range of forecasting is significant and has a substantial socio-economic impact, as many management decisions regarding food security, agriculture, and risk mitigation are within this range, for instance, such forecast would help with the implementation of the necessary disaster mitigation measures that may take two or three weeks (Vitart and Robertson, 2018). Despite the importance of the S2S forecasting for societal applications, however, providing a skilful S2S predictions is challenging. And to enhance the understanding and

skills of forecasting for the S2S range of time, the World Weather Research Programme (WWRP) and the World Climate Research Programme (WCRP) started a five-year project named S2S project in 2013 (Vitart et al., 2012).

An extensive database consisting of S2S model forecasts was created as part of the S2S project. 11 numerical weather prediction (NWP) centres contribute to the S2S project database, these centres provide S2S ensemble predictions using a minimum of four ensemble members and a maximum of 51 ensemble members (Vitart et al., 2017). The S2S precipitation

predictions have a significant number of forecast biases, and the level of predictive skills is marginal (Baker et al., 2019; King et al., 2020). After two weeks, the S2S precipitation predictions' ability to anticipate the occurrence of weekly extreme events is limited (Zhang et al., 2021). Statistical post-processing methods are used to improve the forecasting skills and to remove the forecast biases. Some studies showed that the use of these statistical post-processing methods didn't significantly improve the S2S forecasting skills (Manzanas et al., 2018; Baker et al., 2020).

Many studies showed that Machine learning and deep learning techniques can be successfully used as alternative post-processing tools to improve the precipitation forecast, for instance, Pan et al. (2019) used the convolutional neural network (CNN) to improve the daily precipitation predictions over the contiguous united states, Pan et al. (2021) used the Generative adversarial network (GAN) to correct the daily precipitation climate projection biases over the contiguous united states.



Miao et al. (2019) used a combination of the Convolutional neural network (CNN) and LSTM to improve the prediction of the monsoon precipitations. Wang et al. (2021) used deep learning for the downscaling of daily precipitation and temperature.

The provision of the extensive S2S database increases the potential of using ML for improving the S2S forecast. And to foster the use of ML and AI to improve the S2S ensemble prediction, the world meteorological organisation (WMO), together with the S2S project coordinators, launched an open challenge for using artificial intelligence (AI) or machine learning (ML) to improve the current probabilistic precipitation and temperature S2S forecasts (WMO et al., 2021). The challenge aimed at providing a better biweekly (weeks 3-4 and 5-6 lead times) probabilistic precipitation and temperature forecasts in comparison to the ECWMF hindcast and climatology. A global domain was used with spatial resolution of 1.5 by 1.5 degrees. The main data provided for the challenge consisted of ECMWF real-time forecasts for the year 2020 and reforecasts (hindcasts) for 20 year (from 2000 to 2019) for precipitation and temperature, in addition to precipitation and temperature observations for the same forecast periods provided by the National Oceanic and Atmospheric Administration, Climate Prediction Centre (NOAA, CPC). The participants had to train and validate their models based on the 2000 – 2019 ECMWF 11 ensemble hindcasts and the CPC observations, and to submit their models forecasts for the year 2020, the verification was based on the RPSS score on four domains (Vitart et al., 2022). An appropriate technical environment was provided for the participants, for example, Pinault et al. (2022) provided a S2S predictions data pipeline for machine learning. The top three submission for the competition provided skilful probabilistic forecasts of precipitation and temperature in many regions across the global domain, these submissions were for the CRIMS2S, BSC and UConn teams respectively (Vitart et al., 2022).

CRIMS2S team used a CNN-based weighted average of five models: 1) A post-processed ECMWF, ECCC, and NCEP forecasts, 2) a CNN-based corrected prediction the CNN was applied to the ECMWF forecasts, 3) climatology. The BSC team used the best prediction from four models: 1) climatology, 2) raw ECMWF forecasts, 3) logistic regression (LR), 4) random forest (RF), UCom team divided the global area into 23 regions, a RF classifier is used to improve the forecast, different features were used as input including the statistical parameters of the observations, El Niño–Southern Oscillation (ENSO) indices and past observations (Vitart et al., 2022).

This study investigated the use of different machine techniques and model structures for improving the S2S ECMWF precipitation forecast using the S2S challenge datasets, the evaluation criteria is tailored to be done over selected regions instead of the whole globe to allow for the investigation of as many techniques as possible, the study investigated the use of different individual ML models such as K-NN, LR, ANN-MLP, RF and LSTM where the average ECMWF ensemble hindcasts were used as input, the study also investigated the use of a committee model where a member-by-member correction of the ECMWF ensemble forecast is used, the main objective of the committee model is to preserve the physical relations and inter-dependencies in each ensemble member to allow ML models to better capture these relations and inter-



dependencies, the member-by-member ensemble post-processing was also recommended by Van Schaeybroeck and Vannitsem (2015) and Schefzik (2017).

## 2   Methodology

The categorical classification correction models were built using machine learning (ML) to correct the categorical classifications (above normal, near normal and below normal) of the ECMWF S2S precipitation forecast. Various input datasets were used (four datasets) following spatial and temporal correlation analysis (autocorrelation, cross-correlation and semivariogram). The target output for the categorical classification models was the CPC observed categorical class (above normal, near normal and below normal), which was calculated using the 0.67 and 0.33 quantiles.

The categorical classification correction models were built following two approaches. In the first approach, the ECMWF ensemble mean was used as the input, and various ML techniques were used to build individual models: k-nearest neighbours (k-NN), logistic regression (LR), artificial neural network multilayer perceptron (ANN-MLP), random forest (RF) and long–short-term memory (LSTM). Except for the K-NN, these ML models have the same output layer, which contains three neurons, and the activation function is the SoftMax function, which is used to make the ML models predict the tercile categorical probabilities (above normal, near normal and below normal) at each time step. A schematic representation of how the models in the first approach were built is shown in Fig. 1.

In the second approach, the CM concept was used, in which, instead of using the ECMWF ensemble mean as the input, each realisation or ensemble member (there are 11 ensemble members in the ECMWF hindcast) was trained using a separate ML model and the ensemble output was combined using a logistic regression model. The output layer of the CMs was the same as for the individual ML models with three neurons, and the activation function was the SoftMax function. The CM output was the tercile categorical probabilities (above normal, near normal and below normal) at each time step. A schematic representation of how the models in the second approach were built is shown in Fig. 1.

The models from the first and the second approaches were evaluated in selected assessment regions, which were selected based on the spatial and temporal correlation analysis. The evaluation of the models was done using the RPSS verification metric.





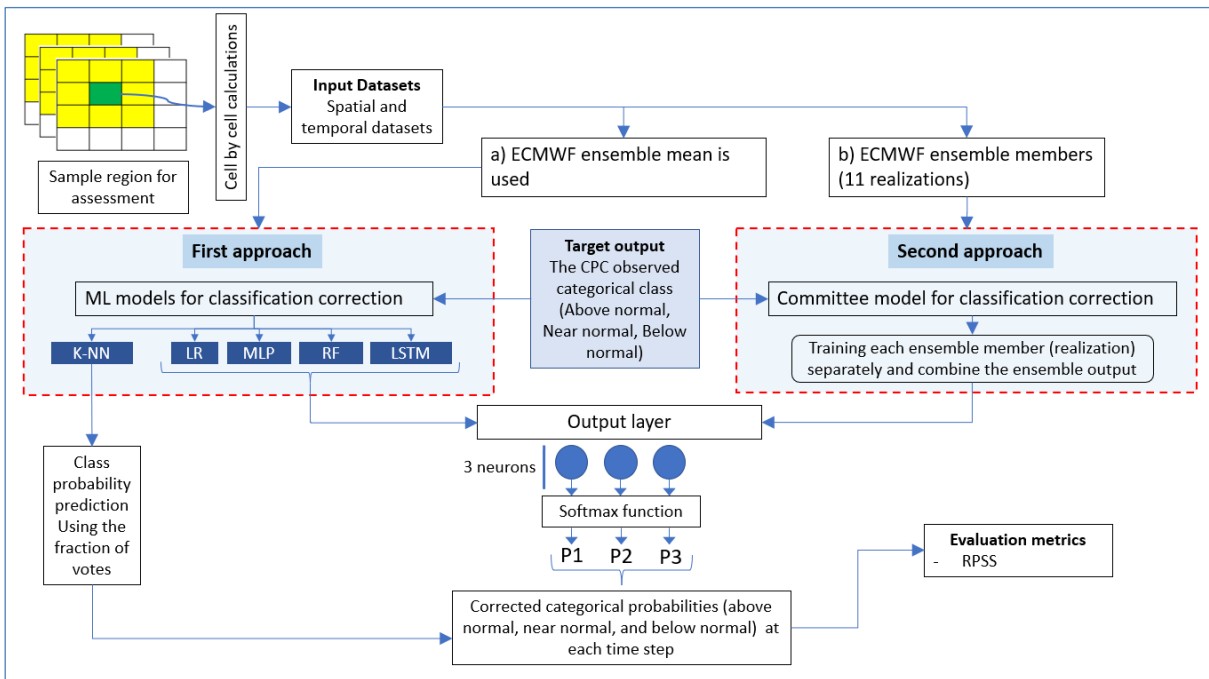

**Figure 1. Schematic representation of how the classification correction models work**

## 2.1 Data

### 2.1.1 ECMWF model data

The meteorological atmosphere is considered a chaotic system on longer time scales (from a few days to weeks); this also depends on the spatial scales. Therefore, in the case of ECMWF, probabilistic forecasting is used to account for uncertainty (ECMWF, 2022).

ECMWF extended-range forecasts are used for the S2S predictions; these forecasts evaluate the predictability in the range between 10 and 46 days. Furthermore, the ECMWF extended-range forecast provided 51 ensemble members for real-time forecasting and 11 ensemble members for hindcast or reforecast (ECMWF, 2021).

### 2.1.2 CPC observations

CPC precipitation observations were constructed using an analysis of gauge-based data consisting of daily precipitation, and the data were collected from more than 30,000 stations from different sources, including the Cooperative Observer Programme (COOP), Global Telecommunications System (GTS) and other national and international organisations (NCAR, 2021).





### 135  2.1.3   S2S AI challenge data

The data provided comprised ECMWF real-time forecasts for 2020, ECMWF hindcasts from 2000 to 2019 and CPC observations from 2000 to 2020. The data were global data consisting of 240 longitudes and 121 latitudes with a spatial resolution of 1.5*1.5 degrees and a forecast time dimension composed of 1060 time-steps (the result of the weekly forecasts for 20 years (1060 = 53 * 20)). At each time step, there were two lead times: first, 14 days lead time, which corresponded to

the aggregation of weeks 3 and 4, and, second, 28 days lead time, which corresponded to the aggregation of weeks 5 and 6. A plot of the lead times for weeks 3 and 4 for ECMWF hindcasts (ensemble mean) and CPC observations are shown in Fig. 2.

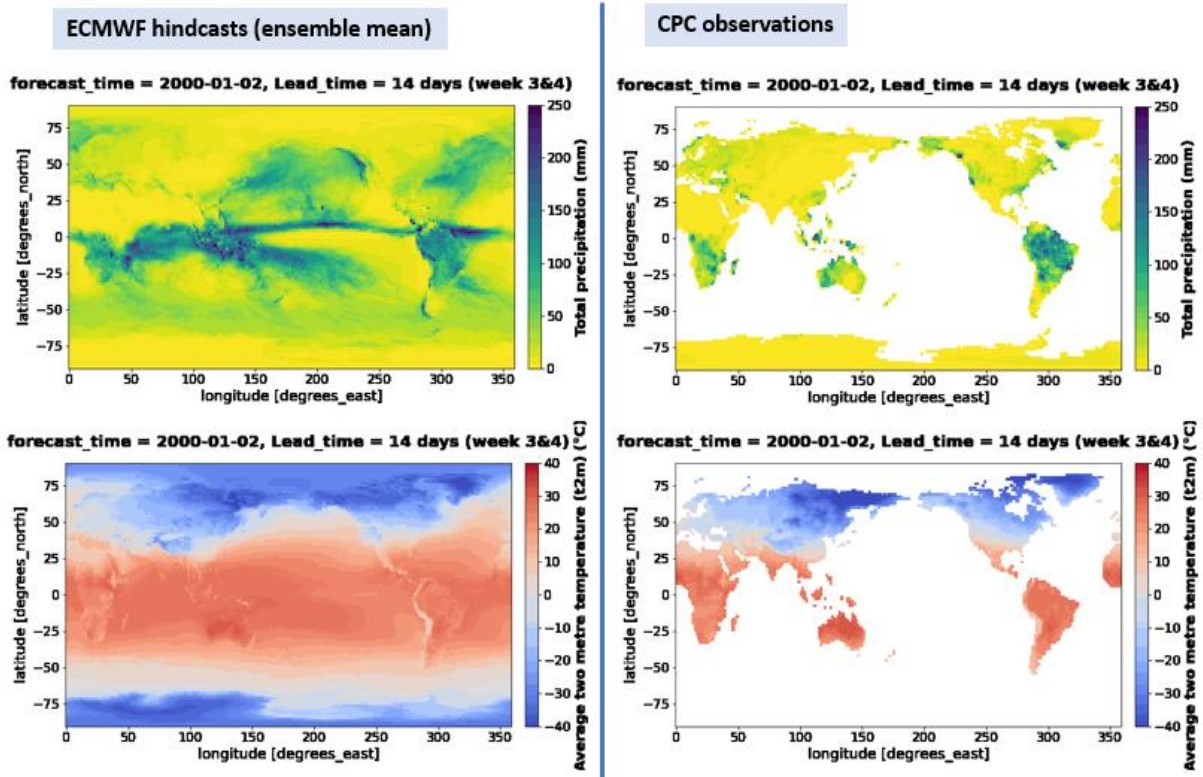

**Figure 2. A single forecast time plot for the tp and the t2m variables using a 14-day lead time (weeks 3 and 4) using the ECMWF hindcasts (ensemble mean) and CPC observations**

### 145  2.2   Spatial and temporal correlation analysis

Spatial and temporal correlation analysis was used for the preparation of temporal and spatial correlated datasets for the training and validation of the machine learning models. The autocorrelation and cross-correlation were measured using the Pearson correlation coefficient (r).





$$r(x,y) = \frac{\sum_{i=1}^{n}(x_i - \bar{x})\ (y_i - \bar{y})}{\sqrt{\sum_{i=1}^{n}(x_i - \bar{x})^2}\ \sqrt{\sum_{i=1}^{n}(y_i - \bar{y})^2}}$$


(1)

Where:

$r(x, y) \equiv$ the Pearson correlation coefficient

$x_i,\ y_i \equiv$ x and y variable values in a sample

$\bar{x},\ \bar{y} \equiv$ mean of the x and y variables

The spatial correlation was measured through the calculation of the experimental semivariogram, which was a function of the difference over a distance that measures the rate of change of variables that vary in space (Olea, 1994; Pyrcz, 2019).

$$\gamma(h) = \frac{1}{2N(h)} \sum_{\alpha=1}^{N(h)} (z(u_\alpha) - z(u_\alpha + h))^2$$

(2)

where $z(u_\alpha)$ is the tail value and $z(u_\alpha + h)$ the head value.

The direct relation between the variogram and covariance function is shown in the following equation:

$$C_x(h) = \sigma_x^2 - \gamma_x(h)$$

(3)

where $C_x(h)$ is the covariance function and $\sigma_x^2$ the variance.

The covariance function is related to the correlogram, as follows:


$$\rho_x(h) = \frac{C_x(h)}{\sigma_x^2}$$

(4)

A comparison between the semivariance and covariance is shown in Fig. 3; the figure also shows three important terms for the variogram interpretation: the sill, nugget and range (Hengl, 2009).




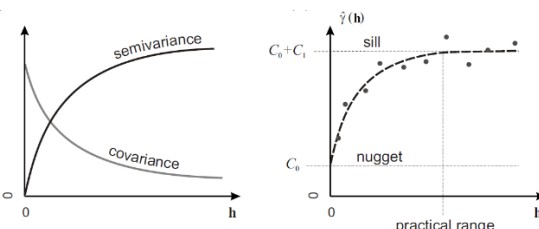

**Figure 3. Basic variogram concepts. Source: Hengl (2009)**

The sill is the variable's variance, the nugget represents either a measurement error or a short-distance spatial discontinuity, and the range is the distance of spatial continuity (Hengl, 2009).

## 2.3     Selection of the assessment regions

The selection of the assessment regions is important because it would create bias to assess the models based on one cell and would be impractical to assess the models based on the entire world because running a machine learning model for the whole world would take a longer time and, in the current study, many machine learning models were investigated.

The selection of the assessment regions was formed in two steps. The first step was the regionalisation of the climatic 180 variables (mainly the total precipitation (tp)) using a cross-correlation analysis, and the second step was the random sample selection per region.

## 2.4     Baseline analysis (ECMWF hindcast)

To evaluate the ECMWF hindcasts, the categorical probabilities were calculated using the 0.67 and 0.33 quantiles, and three categories (above normal, near normal, and below normal) were created from the CPC observations and the ECMWF 185 hindcasts. The 0.67 and 0.33 quantiles were calculated for each week of the year using the biweekly distribution of the (2000 – 2019) CPC observations. The accuracy of the ECWMF hindcasts was evaluated using two verification methods; the first method was the receiver operating characteristic curve (ROC) and the second method the ranked probability skill score (RPSS).

## 2.5     Training and validation of the machine learning models

In general, the ECMWF hindcast data were used for training and cross-validation. Furthermore, the best model in terms of the performance in training and cross-validation was trained and validated using the 28-day lead time (weeks 5 and 6) and tested using the ECMWF real-time forecasting data for 2020 (using the two lead times).





### 2.5.1  K-Nearest Neighbours (K-NN) models

For the K-NN, the CPC observation data were used to build the four models; through trial and error, a value of k equal to 3
was found to be the most appropriate value for providing the best results. The models were tested using the ECMWF
hindcast (ensemble mean) inputs for 2018–2019 (the last two years). For more information about the K-NN model concepts
and algorithms, see annexe (1).

### 2.5.2  Logistic regression (LR) models

For the logistic regression models, the ECMWF hindcast data (ensemble mean) were used for the training and cross-
validation of the models. The first 18 years (02-01-2000 to 31-12-2017) were used for training, and the last two years were
used for cross-validation (02-01-2018 to 31-12-2019). The data were normalised using the min–max transformation method.
In addition, the L2 regularisation term was used to avoid overfitting. For more information about the LR model concepts and
algorithms, see annexe (1).

### 2.5.3  Multilayer perceptron (MLP) models

For the multilayer perceptron, the selection of the training and cross-validation datasets was the same as for logistic
regression. The multilayer perceptron structure is shown in Fig. 4.

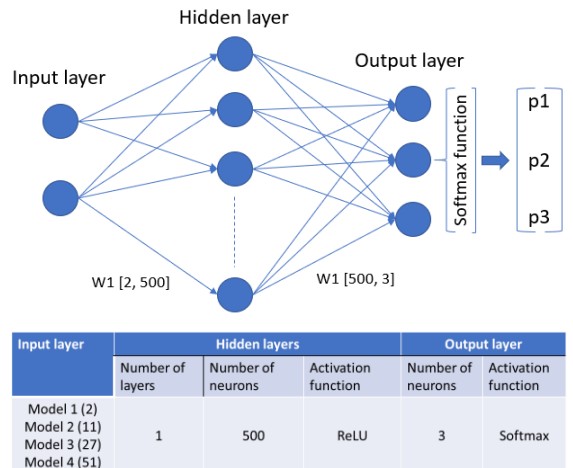

| Input layer | Hidden layers | | | Output layer | |
|---|---|---|---|---|---|
| | Number of layers | Number of neurons | Activation function | Number of neurons | Activation function |
| Model 1 (2) Model 2 (11) Model 3 (27) Model 4 (51) | 1 | 500 | ReLU | 3 | Softmax |

**Figure 4. MLP structures for all models**

To avoid overfitting, the cross-validation data were used, and the training was set to stop when the loss of the cross-
validation data was minimal, ensuring that the MLP would not overfit the training dataset. This is illustrated in Fig. 5.





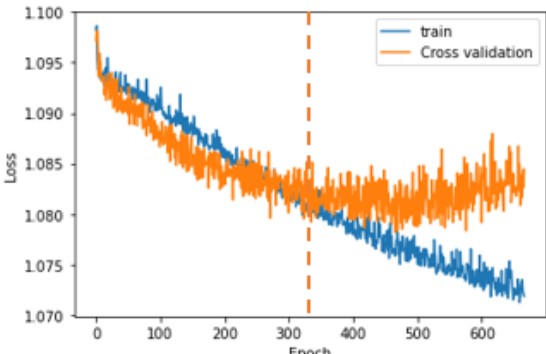

**Figure 5. Loss of training and cross-validation**

As mentioned above and from Figure (24), the training will stop at the dotted line where the loss of the cross-validation is

minimal. To avoid stopping the training in a local minima, a patience term was added to the training. The patience term was

set to 200 epochs so that the training would continue for 200 epochs, which would save the model (checkpoint) at each time

that the cross-validation loss decreased. If the loss did not decrease, then after finishing the 200 epochs, the last saved model

was considered the best (last checkpoint). For more information about the MLP model concepts and algorithms, see annexe

(1).

**2.5.4   Random forest (RF) models**

The training and cross-validation data used for the LR and MLP were used for the RF classifier. The default number of

estimators (the number of trees) was used, which equalled 100 estimators. For more information about the RF model

concepts and algorithms, see annexe (1).

**2.5.5   LSTM models**

The training and cross-validation data used for the LR and MLP were used for the LSTM models. For the training of the

LSTM models, a batch of one-year time series data was used (54 weeks back to the exact date of the last year). The structure

of the LSTM models is shown in Fig. 6. The same procedure used for the MLP models was followed for the LSTM models

to avoid overfitting. For more information about the LSTM model concepts and algorithms, see annexe (1).

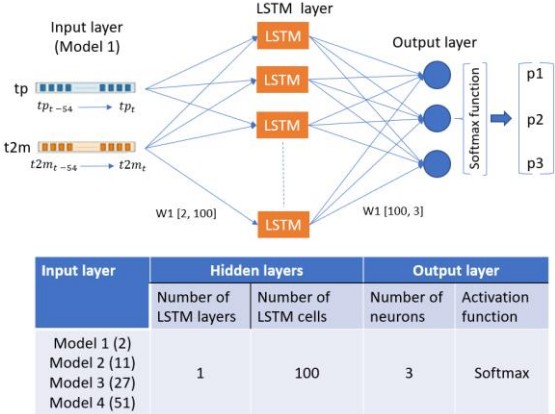

**Figure 6. LSTM models' structure**

### 2.5.6    Committee model (CM)

The structure of the separate MLP models was the same as the structure of the MLP models used for models (1), (2), (3) and (4). In addition, the committee models were built using the data from 2000 to 2018 for training and cross-validation, and the year 2019 was used for testing the committee model using the ECMWF hindcasts data before it was tested using the 2020 ECMWF real-time forecasts. Furthermore, the same four models were built using the committee model structure. The MLP is used to train the separate ML models because it showed slightly better results when the individual ML models were used.

The ECMWF real-time forecasting consisted of 51 ensemble members; for the training of the committee model, only 11 ensemble members were used. Therefore, the first 11 perturbed members from the ECMWF real-time forecasting were used to test the model. For more information about the CM concepts and algorithms, see annexe (1).





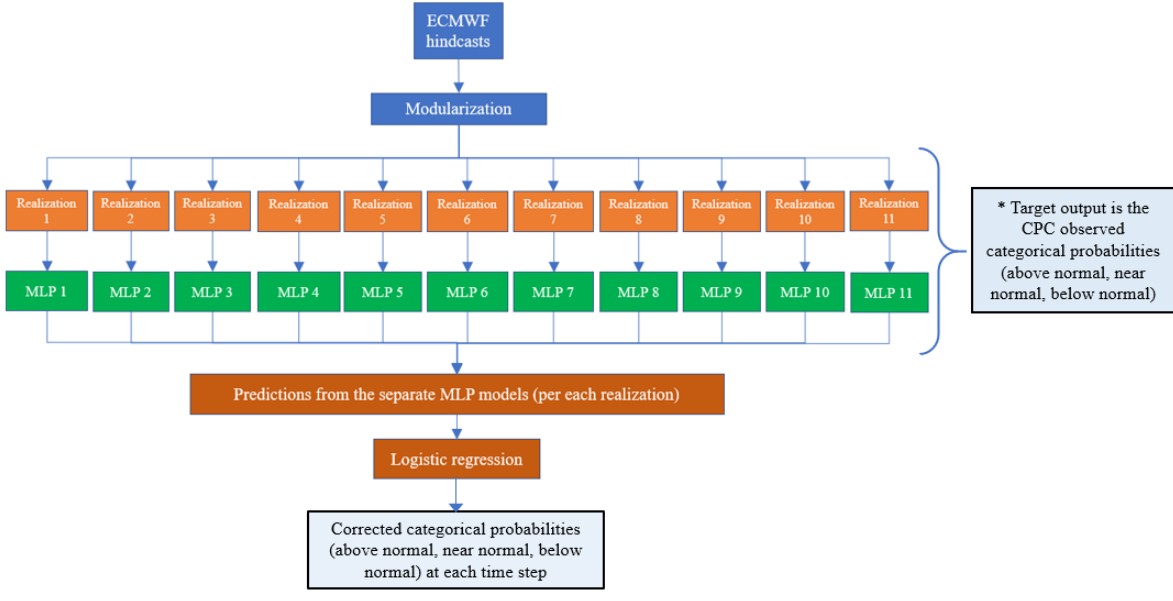


Figure 7. Committee model structure

## 2.6 Verification metrics

Two verification metrics were used for the evaluation of the baseline model (ECMWF hindcast) and the classification correction models (ML models and CMs): the ROC curve and RPSS.

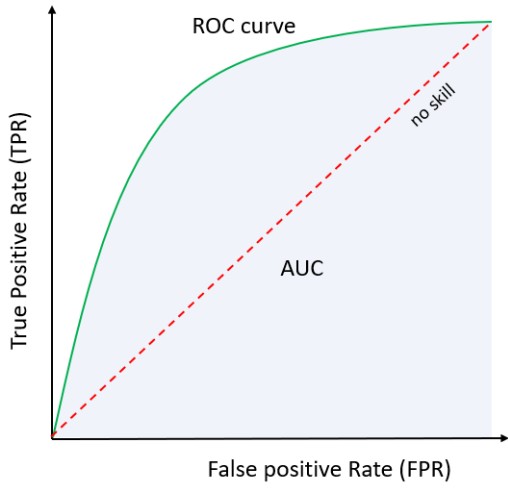


**Figure 8. A demonstrative example of the ROC curve**





The ROC curve measures the forecasting model resolution; it indicates the forecasting model's ability to differentiate between two possible outcomes (an event and a non-event) (WWRP and WGNE, 2021). A demonstrative example is shown in Fig. 8. To plot the ROC curve, the TPR and FPR were calculated from a confusion matrix, which is illustrated in Fig. 9

(Narkhede, 2018).

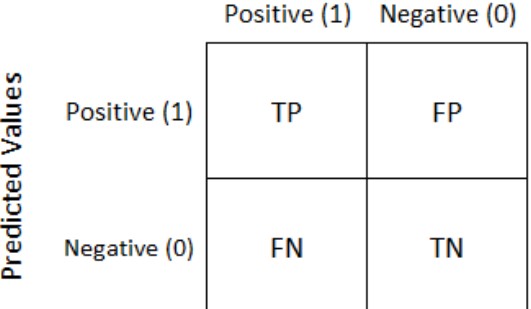

Source: (Narkhede, 2018)

**Figure 9. Confusion matrix**

The TPR and FPR were calculated using the following equations:

$$TPR, Recall \ or \ Sensitivity = \frac{TP}{TP + FN}$$

(5)

$$FPR = 1 - Specificity = \frac{FP}{TN + FP}$$

(6)

The RPSS measures the relative improvement of a categorical probabilistic forecast compared with a reference forecast. The RPSS is calculated using equation (7), where $RPS_{ML \ forecast}$ is the ranked probability score for the ML forecast, $RPS_{ECMWF \ benchmark}$ is the ranked probability score for the reference ECWMF forecasts, the angle brackets $\langle . \rangle$ means that the RPS is averaged over the time dimension period. $RPS_{ML \ forecast}$ and $RPS_{ECMWF \ benchmark}$ can be calculated using equation (8) and equation (9) respectively, where k is the number of probabilistic forecast categories (Weigel et al., 2007):


$$RPSS = 1 - \frac{\langle RPS_{ML \ forecast} \rangle}{\langle RPS_{Reference \ model} \rangle}$$

(7)


$$RPS_{ML\ forecast} = \sum_{k=1}^{k}(ML\ forecast_k - CPC\ observation_k)^2$$

(8)


$$RPS_{Reference} = \sum_{k=1}^{k}(Reference_k - CPC\ observation_k)^2$$

(9)

A positive RPSS value means that the ML model has better forecasting skills than the reference model. Another indication of predictability is to compare the ML model tercile probabilities to a climatological value of one-third (Tippett et al., 2007).

## 3   Results and discussion

### 3.1   Spatial and temporal correlation analysis

The autocorrelation for the total precipitation (tp) variable was calculated, and four lag times (4 time step) were used; the results of the autocorrelation are shown in Fig. 10, for lag = 0, the total precipitation at time = t ($tp_{t=t}$) was compared with the total precipitation at time = t ($tp_{t=t}$), for lag = 1, the total precipitation at time = t ($tp_{t=t}$) was compared with the total 280 precipitation at time = t – 1 ($tp_{t=t-1}$) and so on for the remaining lags. The results of the autocorrelation showed that at lag = 1 still, there was a good correlation in most of the areas that ranged between 0.6 and 1; after lag = 1, the correlation was very low.

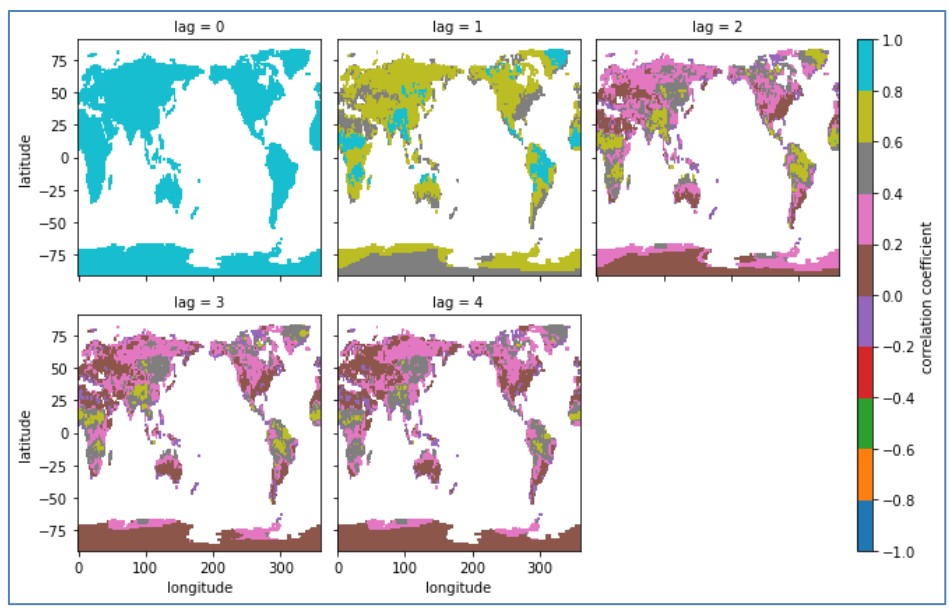

**Figure 10. Autocorrelation for the total precipitation (tp) variable**





The autocorrelation results for the CPC observations were used to determine the selection of the temporal input variables for the machine learning models. From Figure (10), it can be seen that the correlation coefficient for lag equal to 1 was still high for most areas around the world; the correlation ranged between 0.6 and 1 for most of the areas. As a result, the total precipitation at lag = 1 was used as an input for the machine learning models.

    The cross-correlation analysis compared the total precipitation (tp) variable to the two-metre temperature (t2m); the results
are illustrated in Fig. 11, for lag = 0, the total precipitation at time = t ($tp_{t=t}$) was compared with the two-metre temperature at time = t ($tp_{t=t}$), for lag = 1, the total precipitation at time = t ($tp_{t=t}$) was compared with the two-metre temperature at time = t − 1 ($t2m_{t=t-1}$) and so on for the remaining lags. The cross-correlation results showed a very low correlation between the total precipitation and the two-metre temperature in most of the areas for different lag times.

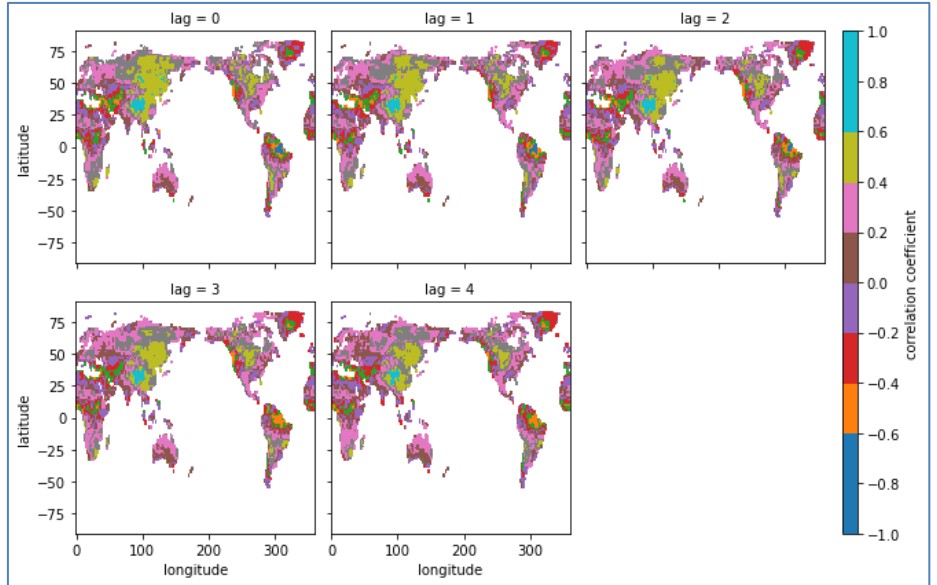

**Figure 11. Cross-correlation between the total precipitation (tp) and the two-metre temperature (t2m).**

    The precipitation data from the ECMWF model was used to calculate the semivariogram because the data were continuous and did not contain nan values over the nonland areas as the CPC observations did (the observation data are provided only over the land areas). The total precipitation (tp) was transformed to a standard normal distribution because, according to the Python library (geostatspy), which was used to calculate the semivariogram, this transformation was required for the
semivariogram simulation, hence providing more interpretable semivariograms.





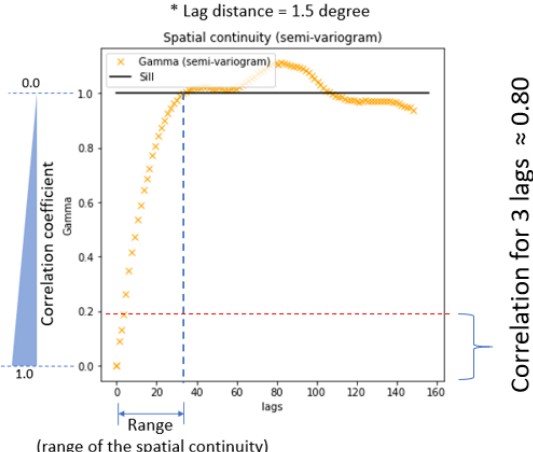

**Figure 12. Spatial continuity (semivariogram)**

The semivariogram is shown in Fig. 8, in which the sill is the variance from the transformed precipitation data and, for the standard normal, the variance is equal to 1. This ensures that there will be a direct inverse relationship between the semivariogram (gamma ($\gamma$)) and correlogram. For instance, when the gamma value is equal to 0, the correlation coefficient is equal to 1 and at the sill, the gamma is equal to 1, and the correlation coefficient is equal to 0.

For the input variable selection and in addition to the inputs from the temporal correlation, the inputs from the spatial correlation were determined using a semivariogram. The correlation coefficient for up to three lags was roughly equal to 0.8. The plot shows fluctuations at very large distance lags, these fluctuations show very small negative and positive correlations. And these fluctuations don't indicate any significant spatial trend. As a result, the spatial precipitation data of up to three lag distances were considered valuable for the training of the machine learning models. The prepared spatial and temporal datasets used for the training and validation of the ML models and CMs are shown in Table 1.







**Table 1. Spatial and temporal input datasets**

| Input dataset 1 | Input dataset 2 | Input dataset 3 | Input dataset 4 |
|---|---|---|---|
| $tp_{t_{ECMWF}}$ <br> + <br> $t2m_{t_{ECMWF}}$ | $tp_{t_{ECMWF}}$ <br> + <br> $tp_{t-1_{ECMWF}}$ <br> + <br> $t2m_{t_{ECMWF}}$ <br> + <br> $tp_{t(x,\ y)_{ECMWF}}$ | $tp_{t_{ECMWF}}$ <br> + <br> $tp_{t-1_{ECMWF}}$ <br> + <br> $t2m_{t_{ECMWF}}$ <br> + <br> $tp_{t(x,\ y)_{ECMWF}}$ | $tp_{t_{ECMWF}}$ <br> + <br> $tp_{t-1_{ECMWF}}$ <br> + <br> $t2m_{t_{ECMWF}}$ <br> + <br> $tp_{t(x,\ y)_{ECMWF}}$ |
| **Number of inputs per dataset** | | | |
| 2 | 11 | 27 | 51 |

## 3.2 Selection of the assessment regions

To select the assessment regions, the results from the cross-correlation were used to create three masks; in the first mask, the correlation range between 0.6 and 1 was used. For the second mask, the correlation range between 0.4 and 0.6 was used, and

for the third mask, the correlation was <= 0.4, as shown in Figures (13), (14) and (15), respectively.





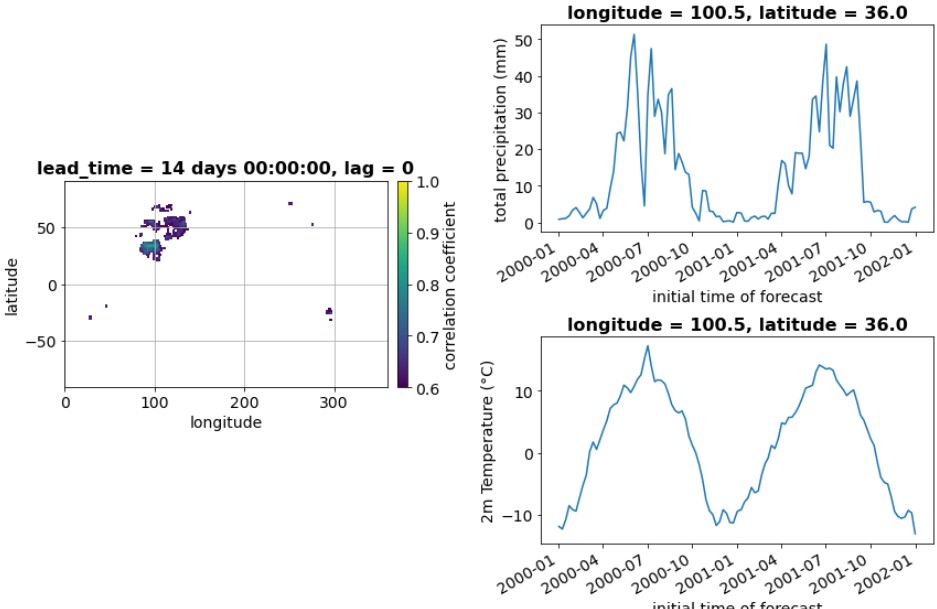

**Figure 13. First mask (cross-correlation range between 0.6 and 1)**

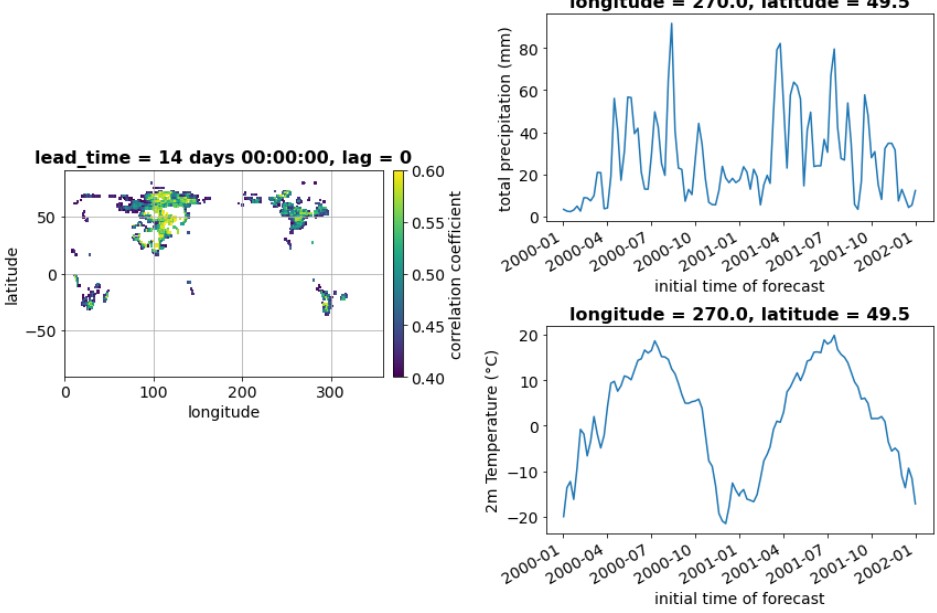

**Figure 14. Second mask (cross-correlation range between 0.4 and 0.6)**





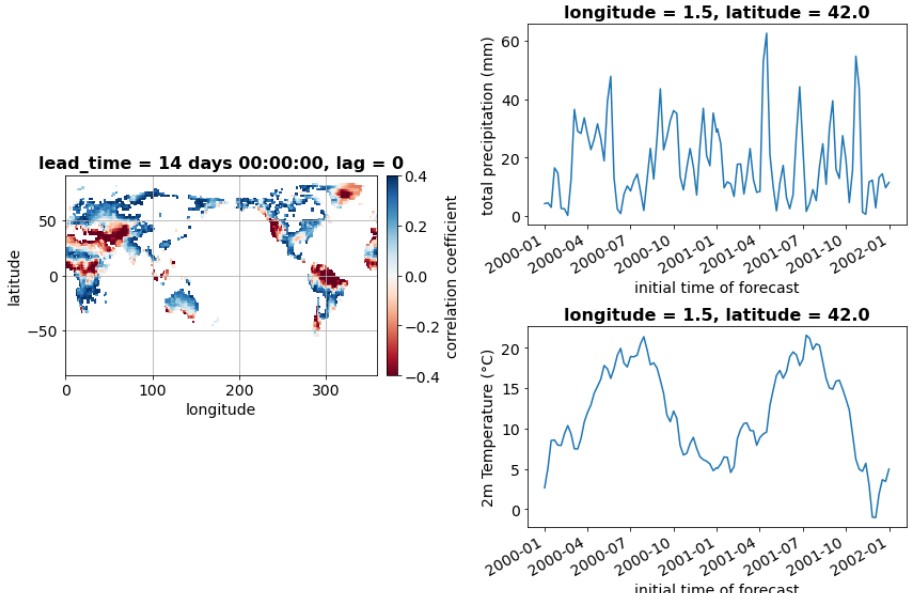

**Figure 15. Third mask (cross-correlation <= 0.4)**

From the first mask (Fig. 13), it can be seen in the time series plot that the correlation was high; in addition, the figure also
shows that the first mask represented a very small area of the whole world. The second mask (Fig. 14) shows that the
precipitation started to fluctuate; in addition, the second mask represented a larger area than the first mask. The third mask
(Fig 15) shows that the precipitation fluctuated even more; in addition, the third mask represented the biggest area of the
world compared with the first and second masks.

Hence, each of these masks represented a different precipitation pattern, so the models in a selected region (sample) from
each of these masks (this would better show which model is performing well for different precipitation patterns) were
evaluated. The selected regions are shown in Fig. 16.





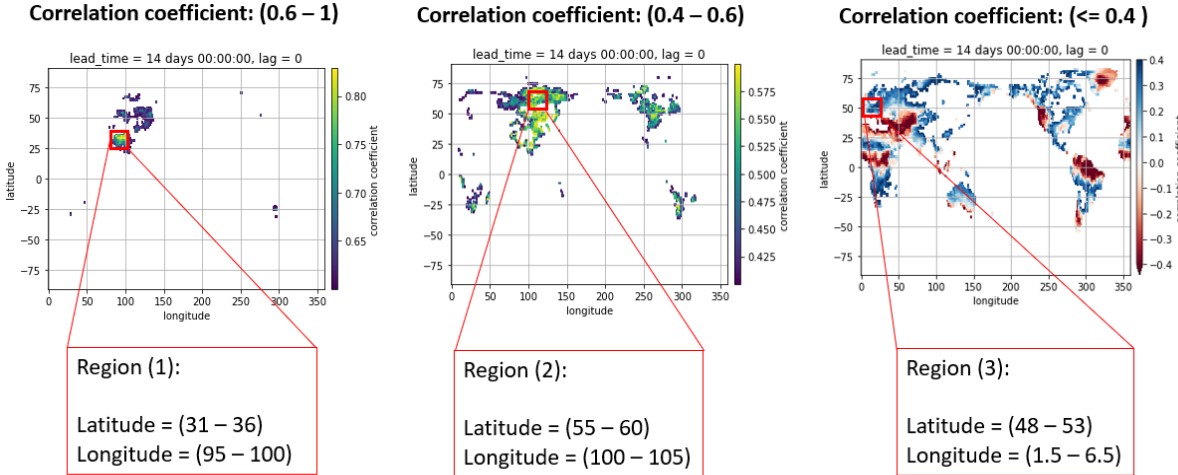

**Figure 16. Selected assessment regions**

### 3.3 Baseline analysis (ECMWF hindcasts)

To evaluate the ECMWF hindcasts, categorical probabilities were calculated using the 0.67 and 0.33 quantiles, and three categories (above normal, near normal and below normal) were created from the CPC observation and the ECMWF hindcasts. For the CPC observations, an illustration for the calculation of the categorical probabilities is shown in Fig. 17. The calculated categorical probabilities were either 0 or 1. Because the CPC observations were provided as single deterministic values at each forecast time step.

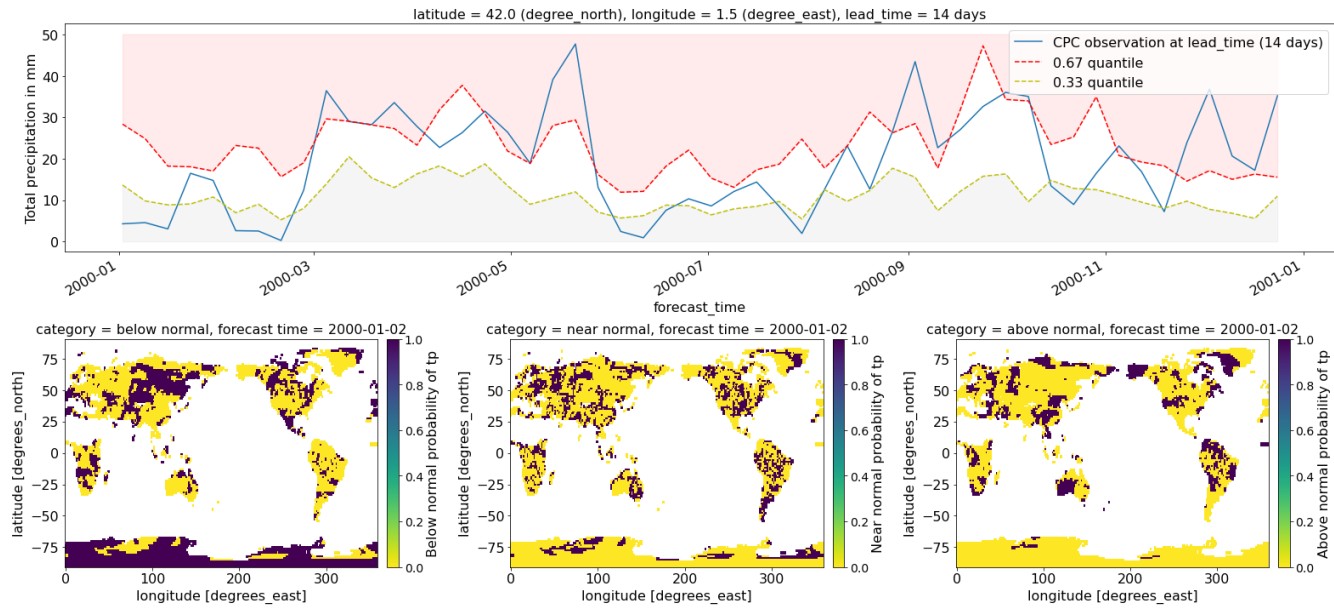


**Figure 17. Categorical probabilities using CPC observations**





On the other hand, the ECMWF hindcasts had 11 ensemble members. The categorical probabilities were calculated as the fraction of ensemble members within each category, here resulting in a categorical probability for each category that varied between 0 and 1. An illustration for the calculation of the categorical probabilities is shown in Fig. 18.


**Figure 18. Categorical probabilities using ECMWF hindcasts**

The categories created from the ECMWF hindcast were compared with those from the CPC observations in the selected assessment regions using the verification metrics (ROC curve and RPSS). The ROC curve results are only shown for one cell

per assessment region for all the evaluated models, either the ECMWF hindcast or machine learning models. For region 1, the cell with 36 latitudes and 100.5 longitudes was selected; for region 2, the cell with 55.5 latitudes and 100.5 longitudes was selected; and for region 3, the cell with 52.5 latitudes and 6 longitudes was selected.




A summary of the evaluation of the accuracy of the weeks 3 and 4 and weeks 5 and 6 ECMWF hindcasts is represented in Tables (2) and (3), respectively. A negative RPSS with respect to climatology was obtained in the three regions for the two lead times (weeks (3 and 4) and weeks (5 and 6)).

**Table 2. Summary for the evaluation of the ECMWF hindcasts accuracy for a 14-day lead time (weeks 3 and 4)**

| Region | Forecasting model | ROC (AUC) | | | RPSS with respect to climatology |
|---|---|---|---|---|---|
| | | above normal | near normal | below normal | |
| Region 1 | ECMWF | 0.562 | 0.501 | 0.563 | -0.517 |
| Region 2 | ECMWF | 0.533 | 0.5 | 0.528 | -0.244 |
| Region 3 | ECMWF | 0.528 | 0.494 | 0.544 | -0.147 |

**Table 3. Summary for the evaluation of the ECMWF hindcasts accuracy for a 28-day lead time (weeks 5 and 6)**

| Region | Forecasting model | ROC (AUC) | | | RPSS with respect to climatology |
|---|---|---|---|---|---|
| | | above normal | near normal | below normal | |
| Region 1 | ECMWF | 0.510 | 0.482 | 0.542 | -0.54 |
| Region 2 | ECMWF | 0.492 | 0.505 | 0.494 | -0.241 |
| Region 3 | ECMWF | 0.507 | 0.490 | 0.509 | -0.165 |





### 3.3.1 ML models results

The cross-validation results for the individual ML models are shown in Table (4).


**Table 4. Model results for lead time = 14 days (weeks 3 and 4)**

| Region | ML model | Model (1) RPSS with respect to | | Model (2) RPSS with respect to | | Model (3) RPSS with respect to | | Model (4) RPSS with respect to | |
|---|---|---|---|---|---|---|---|---|---|
| | | climatology | ECMWF | climatology | ECMWF | climatology | ECMWF | climatology | ECMWF |
| Region 1 | K-NN | -0.251 | -0.3 | -0.072 | -0.11 | -0.083 | -0.127 | -0.193 | -0.247 |
| | LR | -0.021 | -0.067 | -0.014 | -0.06 | -0.017 | -0.063 | -0.017 | -0.063 |
| | MLP | -0.019 | -0.066 | -0.001 | -0.049 | -0.034 | -0.076 | -0.016 | -0.059 |
| | RF | -0.2 | -0.26 | -0.051 | -0.098 | -0.032 | -0.078 | -0.027 | -0.073 |
| | LSTM | -0.03 | -0.078 | -0.049 | -0.099 | -0.09 | -0.137 | -0.05 | -0.10 |
| Region 2 | K-NN | -0.888 | -0.267 | -0.77 | -0.192 | -0.810 | -0.217 | -0.797 | -0.21 |
| | LR | -0.018 | **0.308** | -0.021 | **0.306** | -0.025 | **0.303** | -0.022 | **0.305** |
| | MLP | -0.013 | **0.313** | -0.002 | **0.318** | **0.0057** | **0.323** | -0.002 | **0.318** |
| | RF | -0.14 | **0.23** | -0.029 | **0.302** | -0.026 | **0.3** | -0.031 | **0.299** |
| | LSTM | -0.089 | **0.26** | -0.17 | **0.20** | -0.147 | **0.22** | -0.23 | **0.165** |
| Region 3 | K-NN | -0.531 | -0.369 | -0.492 | -0.334 | -0.504 | -0.346 | -0.475 | -0.32 |
| | LR | -0.0258 | **0.0815** | -0.028 | **0.08** | -0.038 | **0.07** | -0.077 | **0.036** |
| | MLP | -0.028 | **0.08** | -0.034 | **0.075** | -0.019 | **0.089** | -0.008 | **0.096** |
| | RF | -0.2 | -0.07 | -0.07 | **0.04** | -0.051 | **0.059** | -0.045 | **0.064** |
| | LSTM | -0.065 | **0.046** | -0.211 | -0.083 | -0.11 | **0.007** | -0.078 | **0.029** |

The results from model 1 showed that, with respect to climatology, the RPSS was negative for all the models for the three regions, whereas positive RPSS results with respect to the ECMWF hindcast were obtained in region 2 for the LR, MLP, LSTM and RF models and for region 3 for the LR, MLP and LSTM models. For model 1 and the three regions, the MLP and LR were the best machine learning models in terms of the RPSS values.

The results from model 2 showed slight to substantial improvements in most machine learning models' results. For instance, the K-NN and RF model results showed a substantial increase in the RPSS values with respect to climatology and to the





ECMWF for the three regions. Also, the MLP models showed a substantial increase in the RPSS values in regions 1 and 2, but the model did not improve in region 3. In addition, the LR model results showed a slight improvement in the RPSS values in region 1, while in regions 2 and 3, the RPSS results did not improve. The LSTM model results showed a slight

decrease in the RPSS values with respect to climatology and the ECMWF hindcast in the three regions. For model 2, the MLP model was still the best in terms of the RPSS values.

For model 3, RF was the only model that showed an improvement in the RPSS values in all three regions. The MLP models showed a positive RPSS value regarding climatology in region 2 (this is the first time a positive RPSS value was obtained with respect to climatology). The results from model 4 did not show any substantial improvements in all machine learning

models.

The performance of the individual models is considered non-satisfactory as the models failed to obtain positive forecasting skills with respect to climatology, although the adding of the correlated inputs has increased the forecasting skills but not to the extent that positive forecasting skills were obtained. We referred the poor results to either the chaotic behaviour of the model or to the use of the mean ensemble hindcast or both. we thought that the use of the ensemble mean has contributed to

the loss of physical relation and it became difficult for the ML techniques to learn from it.







### 3.3.2 CM results:

The results for the committee models for the three regions are shown in Table (5).

**Table 5. Committee model results**

| Region | ML model | Cross validation | | | | Testing (ECMWF hindcast 2019) | | | | Testing (ECMWF real-time forecast of 2020) | |
|---|---|---|---|---|---|---|---|---|---|---|---|
| | | Week (3&4) | | Week (5&6) | | Week (3&4) | | Week (5&6) | | Week (3&4) | Week (5&6) |
| | | RPSS with respect to | | RPSS with respect to | | RPSS with respect to | | RPSS with respect to | | RPSS with respect to climatology | |
| | | climatology | ECMWF | climatology | ECMWF | climatology | ECMWF | climatology | ECMWF | | |
| Region 1 | Model 1 | 0.21 | 0.18 | 0.22 | 0.2 | 0.154 | -0.038 | 0.1609 | 0.062 | 0.041 | 0.025 |
| | Model 2 | 0.28 | 0.25 | 0.30 | 0.295 | 0.128 | -0.07 | 0.1508 | 0.054 | 0.016 | 0.0042 |
| | Model 3 | 0.28 | 0.25 | 0.31 | 0.30 | 0.119 | -0.081 | 0.138 | 0.037 | -0.0027 | -0.0198 |
| | Model 4 | 0.25 | 0.22 | 0.33 | 0.32 | 0.129 | -0.07 | 0.119 | 0.016 | -0.0062 | -0.0103 |
| Region 2 | Model 1 | 0.069 | 0.36 | 0.069 | 0.35 | 0.0118 | 0.301 | 0.0106 | 0.238 | 0.0018 | 0.014 |
| | Model 2 | 0.12 | 0.4 | 0.111 | 0.374 | 0.0066 | 0.297 | 0.0014 | 0.232 | -0.016 | -0.021 |
| | Model 3 | 0.12 | 0.4 | 0.143 | 0.398 | 0.01 | 0.3 | -0.0169 | 0.218 | 0.00047 | -0.0158 |
| | Model 4 | 0.128 | 0.41 | 0.148 | 0.40 | 0.008 | 0.2985 | -0.044 | 0.194 | -0.0098 | -0.0247 |
| Region 3 | Model 1 | 0.083 | 0.18 | 0.089 | 0.18 | -0.0093 | 0.1292 | 0.00296 | 0.1029 | 0.0262 | 0.024 |
| | Model 2 | 0.11 | 0.2 | 0.146 | 0.23 | -0.015 | 0.124 | 0.006 | 0.106 | 0.013 | -0.004 |
| | Model 3 | 0.11 | 0.2 | 0.161 | 0.245 | -0.059 | 0.086 | -0.0313 | 0.0717 | -0.0495 | -0.049 |
| | Model 4 | 0.141 | 0.23 | 0.169 | 0.25 | -0.0433 | 0.0994 | -0.031 | 0.072 | -0.031 | -0.043 |

The committee models showed very interesting results in general. The cross-validation results (weeks 3 and 4 and weeks 5 and 6) showed positive RPSS values regarding climatology and ECMWF in all the models in the three assessment regions. As previously noted, model 2 showed a substantial increase in the RPSS values in all three regions. In addition, models 3 and 4 did not show any substantial improvements in the RPSS values.

The testing results (weeks 3 and 4 and weeks 5 and 6) using the 2019 ECMWF hindcast were very good. Positive RPSS values with respect to ECMWF were obtained for the four models in three regions; also, positive RPSS values with respect to climatology were obtained in almost all models.



The testing results (weeks 3 and 4 and weeks 5 and 6) using 2020 real-time forecasts showed that model 1 had the best performance, where the RPSS values with respect to climatology were positive for all the assessment regions. The performance decreased in models 2, 3 and 4.

The committee model results can't not be directly compared to the results in the S2S competition because the evaluation criteria is a bit different, however, the comparison can give an indication about the performance of the CM in comparison to the top submissions, the CM results are within the best three submissions. In addition, the CM results were obtained by using only the raw ECMWF ensemble hindcasts/forecasts as inputs while different inputs were used by the rewarded teams such as the climatology, which for example for the BSC team prevents the model outcome to be worse than climatology.

Furthermore, the cross-validation and testing results of the ECMWF hindcasts showed promising results and it is expected that with fine-tuning and the addition of more input features that can improve the learning process such as the Madden-Julian Oscillation (MJO) which is considered one of the sources of predictability for this range of forecast, a better, and more accurate results for cross-validation and testing can be achieved.

## 4    Summary and conclusion

Adding temporally and spatially correlated inputs substantially increased the accuracy of the machine learning models. This can be seen in results from the K-NN in model 2. Because the K-NN is a nonparametric model, its accuracy essentially depends on the selection of the input variables.

In models 3 and 4, the remaining spatially correlated inputs were added; the results from models 3 and 4 showed either a slight increase in the accuracy or slight decrease in the accuracy; however, there were no substantial improvements in the

models' results.

The MLP and LR models had the best performances among the other machine learning techniques. Still, the aim of getting a positive RPSS with respect to climatology was not fulfilled; the only positive RPSS value with respect to climatology was 0.0057 in region 2 for model 3 using MLP.

The committee model was used with the aim of preserving the physical relation through the training of each realisation

separately; the results from the committee model were interesting because the cross-validation results showed positive RPSS values with respect to climatology in the three regions for all four models.

Because the committee model showed the best results compared with other machine learning models, the committee model was also used for the classification correction of weeks 5 and 6 (28-day lead time).

For the 14- and 28-day lead times, model 2 inputs have shown a substantial increase in the committee model results in the

three assessment regions, whereas in models 3 and 4, a slight increase in the models' performances over model 1 was obtained.

The committee model was tested for the 2019 ECMWF hindcasts; the results were very close to the cross-validation results and were better than any of the machine learning models that used the ECMWF ensemble mean.

The committee model was also tested using the real-time forecasting of 2020 for weeks 3 and 4 and weeks 5 and 6. Because real-time forecasting provides 51 ensemble members, only the first 11 perturbed members were used for testing. The results for weeks 3 and 4 and weeks 5 and 6 showed that model 1 had the best RPSS results with respect to climatology. The performance decreased for models 2, 3 and 4.

As a conclusion of our research, we found that the concept of committee model (CM) is a promising approach that can be further studied and evaluated using different combination of the state-of-the-art ML techniques to maximise its potential in improving the S2S ensemble precipitation forecast.

## 5 Data and code availability

The AI challenge data is publicly available on Renku platform which is a repository that is developed by the Swiss Data Science Centre (SDSC), the data accompanied with a useful information about the AI challenge can be found within a project template named S2S-ai-challenge-template and it can be easily accessed using the link in the reference (Spring et al, 2022).

All the main codes that are developed and used to conduct this research are available in GitHub repository, the codes can be accessed using the digital object identifier (DOI) in the reference (Elbasheer et al, 2022).

## 6 Author contribution

MEEEE, GAC, and DS were responsible for the conceptualization and methodology development, MEEEE and GAC carried out the formal analysis and investigation, MEEE was responsible for software and code development, DS was responsible for research supervision, EAV carried out revision and validation, MEEE and GAC carried out the original draft preparation and MEEE, GAC, DS and EAV carried out review & editing.

## 7 Competing interests

A co-author is a member of the editorial board of HESS journal. The authors have no other competing interests to declare.

## 8 Acknowledgements

This work was based on S2S data. S2S is a joint initiative of the World Weather Research Programme (WWRP) and the World Climate Research Programme (WCRP). The original S2S database is hosted at ECMWF as an extension of the TIGGE database.





The current study was carried out as part of the Climate Intelligence (CLINT) project. The main objective of the project is
the use of AI to improve climate science, services and information systems. The project is funded by the European Union
(EU) and started in July 2021.










# 9    Appendix A

## 9.1    Machine learning techniques

### 9.1.1    K-nearest neighbours (K-NN)

The K-NN algorithm is considered a nonparametric classification technique. In K-NN, an uncategorised sample is classified
based on the classification of its neighbours. The uncategorized sample is assigned a class, which is the predominant class of
the k nearest neighbours. To determine the nearest neighbours, the Euclidean distances between the data points were
calculated (Mehdizadeh, 2020).

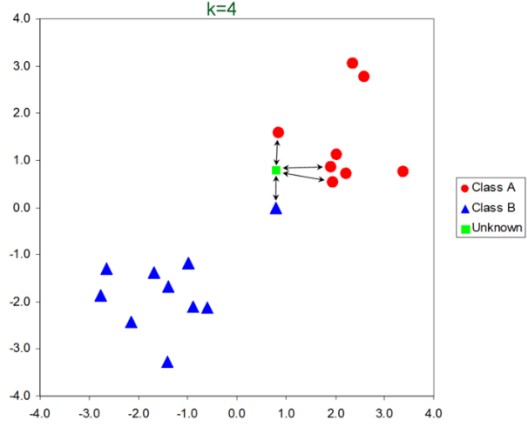

*Source: (Peterson, 2009)*

**Figure A1. Example representation of the K-NN; when four nearest neighbours are
used, the predominant class is the red colour class, and the red class is assigned to the
unknown sample.**


### 9.1.2    Logistic regression

A logistic regression is used when there is a categorical dependent variable. For instance, the model can be defined as a
model that builds a relationship between one or more independent variables and the categorical dependent variable. In
addition, the model relates the independent variables to the categorical variables through a logistic regression equation (Lee
and Kim, 2021).

In a logistic regression, a linear regression is first built from the input variables; then, it is used as an input for a logistic
function to calculate the probability, and for logistic regression, the sigmoid function is used (Lee and Kim, 2021):

$$z = \beta_1 \, x_1 + \beta_1 \, x_2 + \ldots + \beta_n \, x_n$$

(A1)






$$h\Theta(x) = sigmoid(z) = \frac{1}{1 + e^{-z}}$$

(A2)

$h\Theta(x)$ is the probability that the output is equal to 1.

The use of the sigmoid function ensures that, for an independent variable between -∞ and +∞, the output will always be between 0 and 1 (Lee and Kim, 2021; see Figure (A2)).

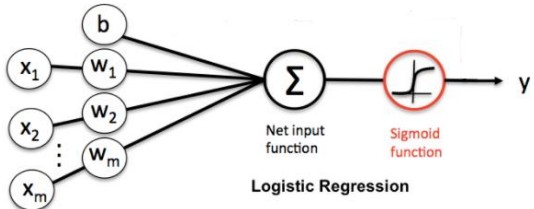


*Source: (KDnuggets, 2021)*

**Figure A2. Softmax regression for binary
classification problem**

For data with more than two output categories or classes, the problem is referred to as multinomial logistic regression, in which the SoftMax function is used instead of the sigmoid function (Goodfellow, et al., 2016; see Fig. A3); the equation of

the SoftMax function is presented below.

$$softmax(z) = \frac{e^{z(i)}}{\sum_{j=0}^{k} e^{z(j)}}$$

(A3)

The length of the input vector z is equal to the number of classes.

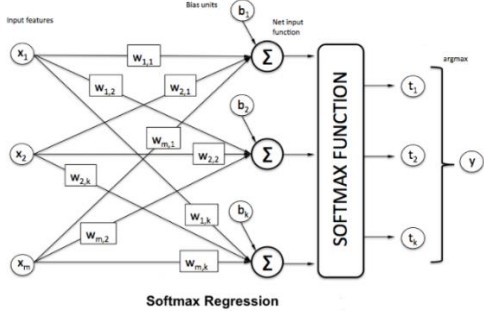

*Source: (KDnuggets, 2021)*


**Figure A3. Logistic regression for multinomial
logistic regression**

To avoid overfitting, two regularisation terms can be used. These regularisation terms are L1 and L2.





The L1 regularisation term is called a Lasso regression. It adds a penalty term to the loss function equal to the absolute value of the magnitude of the coefficients. The loss function and L1 regularisation term are shown in Equation (A4).


$$Loss\ function = \ Loss\ term + \ \lambda \sum_{j=1}^{p} |\beta_j|$$

(A4)

The L2 regularisation term is called a Ridge regression, and it adds a squared magnitude of coefficients to the loss function as a penalty term. This is shown in Equation (A5).

$$Loss\ function = \ Loss\ term + \ \lambda \sum_{j=1}^{p} \beta_j^2$$

560                                                                                                          (A5)

The lambda value should not be very large because this will lead to underfitting. The default value of lambda is 1.

In ML, the input and output data are normalised to ensure efficient learning. Here, the min–max transformation method can be used to transform the data into the range of 0 and 1. This is shown in Equation (A6) below.

$$x' = \frac{x - \min(x)}{\max(x) - \min(x)}$$

565                                                                                                          (A6)

### 9.1.3   Multilayer perceptron (MLP)

Multilayer perceptron (MLP) is considered a feed-forward neural network; it consists of inputs and outputs layers; in addition to hidden layers that can be one or more layers, each hidden layer contains one or more neurons, and each neuron in the hidden layer must have an activation function (Hagen et al., 2021).

Inside the neuron, the weighted sum of the inputs is calculated and used as an input to the activation function. Usually, the sigmoid and rectified linear unit (ReLU) functions are used, but an arbitrary activation function can still be used. The MLP is called a feed-forward neural network because each layer feeds the next from the input layer through the hidden layers to the output layer (Bento, 2021).

The use of ReLU as an activation function in the hidden layers arises from the fact that the function is scale-invariant,
providing better optimisation when the stochastic gradient descent (SDG) is used (Bento, 2021). In addition, it is several times faster in training compared with the other activation functions (Krizhevsky et al., 2012).





The learning mechanism for the MLP is called the error backpropagation, in which an optimisation function is used. The typical optimisation function is the gradient descent. To be able to calculate the gradient descent, the activation and weighted sum functions in the neurone should be differentiable (Marius et al., 2009).

In the backpropagation process, the gradient of the error function is calculated at the end of each feed-forward step, and then, the weights of the neural networks are adjusted with updated values from the gradient. This update of the weights is back-propagated from the output layer to the start of the neural network; this process is iteratively done until the errors are minimal or a convergence threshold has been reached  (Marius et al., 2009).

For categorical classification problems, categorical cross-entropy (CCE) is used as a loss function; the categorical cross-
entropy is shown in equation (A7).

$$CCE = -\sum_{c=1}^{M} observed_c * \log(predicted_c)$$

(A7)

where M is the number of categorical classes, $observed_c$ is the observed probability for the categorical class, and
$predicted_c$ is the predicted probability for the categorical class.

The Adam optimisation algorithm is used as the optimisation function. It combines the properties of the adaptive gradient algorithm (AdaGrad) and the root mean square propagation (RMSProp) algorithms to handle problems with sparse or very noisy gradients (Kingma and Ba, 2014; see Fig. A4).

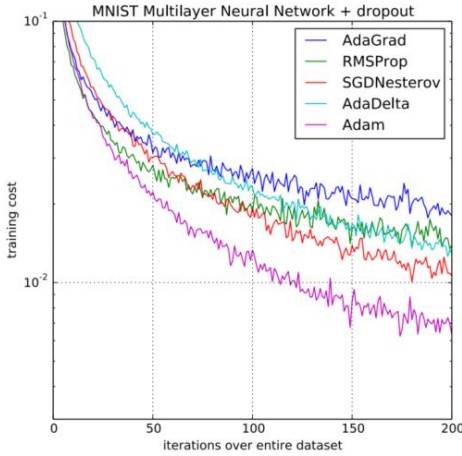


**Figure A4. The performance of Adam algorithm with respect to other optimisation algorithms using multilayer perceptron and large dataset. Source: Kingma and Ba (2014)**





### 9.1.4    Random forest

Random forest is a technique that can be used for classification and regression problems. Random forest is an ensemble of decision trees, and it is immune to overfitting (Breiman, 2001, as cited in Schoppa et al., 2020).

To ensure the diversity of the decision trees, bagging (bootstrap aggregation) can be used. In bagging, small changes are made to the training dataset in each tree to ensure that there are different decision trees.

The splitting of the data in each tree is done through the calculation of the information gain using entropy, which measures the impurity of the sample (S); entropy is illustrated in equation (A8).

$$Entropy(S) = \sum_{i=1}^{c} - p_i \, \log_2 p_i$$

605                                                                                                     (A8)

where c is the number of classes and $p_i$ is the probability of the class.

### 9.1.5    Long–short-term memory network (LSTM)

Architecturally, the difference between the LSTM and traditional recurrent neural networks (RNNs) is in the recurrent cell's internal structure. For instance, the recurrent cell of the traditional RNNs consists internally of one state ($h_t$) (see Figure

(25)). In each time step, this internal state is recomputed using the following equation:

$$h_t = g(W x_t + U h_{t-1} + b)$$

(A9)

G (F(X)) is called the activation function. In the case of the traditional RNNs, the activation function is the hyperbolic tangent (Kratzert et al., 2018).

On the other hand, in an LSTM, there is an additional cell state ($c_t$) that can store the information for a long time; this cell is also called cell memory. In addition to the cell memory, the recurrent cell of the LSTM also contains three gates: the forget, input and output gates (these gates are shown in Fig. A5; Kratzert, et al., 2018).

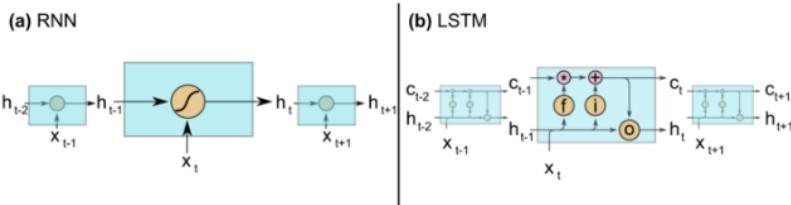

**Figure A5. Difference between the traditional RNN and LSTM.**
**Source: Kratzert et al. (2018)**


The forget gate determines the elements that will be forgotten from the cell state vector ($c_{t-1}$) by using the following equation:





$$f_t = \sigma(W_f x_t + U_f\, h_{t-1} + b_f)$$

(A10)

The activation function for the forget gate is the sigmoid function, where $W_f$, $U_f$ and $b_f$ are learnable parameters.

Then, a potential update vector ($\widetilde{c_t}$) is calculated for the cell state using the last hidden state $h_{t-1}$ and the current input $x_t$.

$$\widetilde{c_t} = tanh(W_{\tilde{c}} x_t + U_{\tilde{c}}\, h_{t-1} + b_{\tilde{c}})$$

(A11)

To update the cell state, the input gate determines which information is used from ($\widetilde{c_t}$)

$$i_t = \sigma(W_i x_t + U_i\, h_{t-1} + b_i)$$

(A12)

$$c_t = f_t \odot c_t + i_t \odot \widetilde{c_t}$$

(A13)

$\odot$ refers to the element-wise multiplication.

The output gate determines which information is used from the updated cell state ($c_t$) to calculate the current hidden state ($h_t$).

$$o_t = \sigma(W_o x_t + U_o\, h_{t-1} + b_o)$$

(A14)

$$h_t = tannh(c_t) \odot o_t$$

(A15)

The current hidden state was used to calculate the prediction.

$$prediction = W_d h_n + b_d$$

(A16)

where $h_n$ is the last LSTM layer output.

**9.1.6    Committee model (CM)**

The CM concept uses a combination of separate neural networks or ML models designed to replicate the same phenomena. The output of these models (ensemble) is combined using simple averaging or weighting method (Haykin, 1999; see also Corzo et al, 2009; Corzo et al, 2007; Corzo, 2009). Figure (A6) illustrates how the CM works.






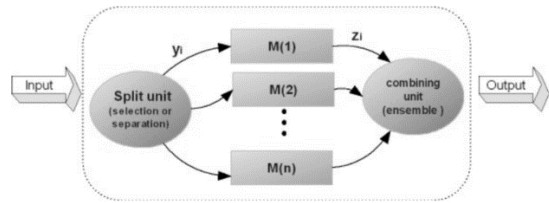

**Figure A6. Committee model structure. Source:
Corzo et al. (2009)**

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
