# Peer review of "Machine Learning and Committee Models for Improving ECMWF Subseasonal to Seasonal (S2S) Precipitation Forecast"

_Hydrology and Earth System Sciences, 2023_

## Referee Comment (RC3)

**Machine Learning and Committee Models for Improving ECMWF Subseasonal to Seasonal (S2S) Precipitation Forecast**

Overall comment:

This paper builds categorical classification correction models to correct the ECMWF S2S precipitation forecast by adopting machine learning and committee models. The result may be meaningful and practical to management decisions regarding food security, agriculture, and risk mitigation.

However, the methodology is routing and the work is not innovative enough, although it considers spatiotemporal information and machine learning algorithms to construct the S2S correction model. In addition, there is still a lot large space for improvement, especially since it is not a scientific paper.

As such, I don't recommend the publication of this manuscript in Hydrology and Earth System Sciences in its present form.

Major comments:

(1) This paper adopts the traditional way to post-process the S2S precipitation forecast by machine learning algorithm, and the machine learning algorithm is not innovative. Besides, the committee model is essentially the same as the stacking integration algorithm, and the stacking integration algorithm may be better than the committee model. On the other hand, S2S was considered a difficult time range for weather forecasting, being both too long for much memory of the atmospheric initial conditions and too short for SST anomalies to be felt sufficiently strongly, making it difficult to beat persistence. Therefore, it is not enough to only consider early precipitation and temperature as model inputs, but also to consider the Madden-Julian Oscillation (MJO), ocean conditions, soil moisture, and so on.

(2) The abstract should have been brief and logical. However, the abstract is redundant, it is recommended to only introduce the core content to make it more logical. For example, there is no need to spend 13 lines introducing the first approach, which reduces the correlation between the two approaches.

(3) In the introduction, it is recommended to remove the methods used in the top three submissions for the competition (lines 83-88), which are not related to the content of the introduction. In addition, it is recommended to explore the state-of-the-art in using machine learning techniques for improving S2S precipitation forecasts and then introduce the innovative aspects of the proposed method compared to existing machine learning methods.

(4) In the methodology, the ECMWF extended-range forecast provided 100 ensemble members for real-time forecasting, so the corresponding content should be modified (lines 237, 465 )

(5) In the methodology, the structure of the separate MLP models was the same as the structure of the MLP models used for models (1), (2), (3) and (4). It's not clear what the models represent. Therefore, it is recommended to add Table 1 to the appropriate location in the methodology.

(6) In the results and discussion, the content of section 3.2 can be transferred to section 2.3.

(7) In the results and discussion, the authors compare the performance of different ML methods on ECWMF and climatology using the CRSS metric, but I don't understand what data climatology represents.

(8) In the results and discussion, the cross-correlation results showed a very low correlation between the total precipitation and the two-metre temperature in most of the areas for different lag times, so why did the authors choose temperature as the input for the models?

(9) In the results and discussion, the authors only briefly described the results. I don't see any further in-depth analysis. For example, models 2, 3, and 4 adopt different spatial information, what conclusions can be drawn from the differences in their results. Unfortunately, similar conclusions have not been seen in the results and discussion. It is recommended that some of the conclusions be added to the results and discussion.

(10) Consistent with the opinions of previous reviewers, conclusions are those of a technical report, with no attempt at explaining what it brings to the state of the art.

Others:
(1) Line 52, there are 12 numerical weather prediction (NWP) centres that contribute to the S2S project database.
(2) Line 106, "k-NN" should be "K-NN".
(3) Line 214, "Figure (24)" does not exist.
(4) Lines 262-263, "$RPSE_{ECMWF\ benchmar}$" is inconsistent with the formula in the paper.